# Measuring Occupational Well-Being Indicators: Scale Construction and Validation

**DOI:** 10.3390/bs14030248

**Published:** 2024-03-19

**Authors:** Hanvedes Daovisan, Ungsinun Intarakamhang

**Affiliations:** Behavioral Science Research Institute, Srinakharinwirot University, Bangkok 10110, Thailand; hanvedes@g.swu.ac.th

**Keywords:** occupational well-being, employee well-being, construct validity, confirmatory factor analysis, Laos

## Abstract

The purpose of this study is to carry out the scale development of occupational well-being (OWB) (affective, professional, social, cognitive, psychological and psychosomatic well-being) in Laos. Using multiple sampling data, we developed a valid OWB scale with a large Laotian sample (*n* = 1745). The validity of the OWB-47 scale was analyzed using exploratory factor analysis (EFA). Cross-validity, the initial model, and confirmatory factor analysis (CFA) were performed using Stata 19 to assess the validity of the scale development. Consistent with the valid model, the CFA revealed a unidimensional structure in the OWB scale. The initial measurement of the OWB scale was significantly correlated with the measure of the six-dimensional model. Regarding the full model testing, the CFA model was developed to test the validity of the OWB-47 scale, suggesting the acceptability of the fit model.

## 1. Introduction

The occupational well-being (OWB) scale of employees is a major research concern within organizational fields [1,2]. As one of the most significant existing constructs, the OWB construct is central to employee improvement efforts [3,4,5]. Scale development and validity are critical to much of the work in employee well-being, work characteristics, and organizational behaviors. Like previous studies, Warr [6,7] drew on three dimensions of Ryff’s [8] OWB: affective, professional and social. According to Sonnentag et al. [9], the OWB construct is defined by three dimensions: cognitive, psychological and psychosomatic well-being. The valid scale is typically used to capture social, professional and organizational well-being that cannot be captured in a single variable or item.

Some studies have developed a single construct of the OWB scale, especially when considering context-free measures of organizational employees [10]. A large number of OWB constructs are different dimensions: self-acceptance, environmental mastery, autonomy, positive relations, personal growth, and purpose in life [7,11,12]. Research on OWB constructs has been valuable, but there is merit in measuring the findings of related organizational fields [13,14,15]. Furthermore, it deliberately aims to encompass multiple domain constructs of OWB indicators: affective, professional, social, cognitive, psychological and psychosomatic well-being [2,16,17,18]. Unfortunately, unlike work characteristics, no measure exists to capture occupational well-being at work from an organizational perspective [2,7,19].

Instead, most scholars have constructed OWB with relatively broad scale development by focusing on employee levels [3,20,21,22]. They have retooled the domain constructs of the OWB scale for employee well-being as a single item (e.g., I am an important part of my team and organization) [23,24]. Although some constructs of OWB scales may indeed be relevant to employees, others may be invalid constructs for measuring organizational perspectives. As previous studies have clarified, OWB constructs are different scales in measurement, which depend on population, organizational employees, and social context [1,4,13,17].

This issue seems to be especially relevant in the context of the OWB scale in South East Asia [25], with the various constructs having no scale that covers the six dimensions of OWB in Laos [26,27,28]. This raises a knowledge gap in how constructs of OWB indicators should be multiple scales: does OWB mainly measure organizational employees with Laotian samples [26,28]? However, the measurement of previous constructs has indicated strong collectivist societies for measuring OWB scales, using Laotian employees [27]. Potentially, multidimensional approaches to measuring OWB constructs consist of affective, professional, social, cognitive, psychological and psychosomatic well-being, which are key contextual constructs identified as important for scale validation in organizational fields [2,3,7,12,23].

OWB constructs of Laotian samples have measured single scale development [26,27,28], which is needed to construct scale validity. First, relative to the construct of global scales, domain-specific measures of OWB remain scarce [3]. The refinement of tools for assessing OWB scales is a crucial part of occupational characteristics, organizational employees, and society [2,6]. Second, previous studies have not assessed multidimensional constructs to measure a six-dimensional OWB scale in Laos [27]. We developed scales that focus mainly on cognitive well-being (psychological, social, subjective and professional well-being) and affective well-being (affective and psychosomatic well-being) [18,29,30,31] in organizational employees.

Many studies have measured OWB constructs using a small sample, such as cross-sectional studies and systematic intervention studies, which involve single scale validity [2,11,12,13,19,22,23]. Given the brevity of the construct in such studies, the availability of organizational soundness would be beneficial to measure the OWB scale. Currently, however, construct scales are single-dimension scales; thus, previous studies have often constructed limitations from organizational employee perspectives. The lack of a valid OWB scale in the current organizational literature exists because employee well-being is limited in its ability to capture what it means to flourish at work especially.

The primary purpose of this study was to develop valid constructs of the OWB scale for Laotian samples. We followed the scale development recommended by Van Horn et al. [12], Hyvönen et al. [19], Khatri and Gupta [23], Churchill [32], and Veldhoven and Meijman [33] to measure OWB constructs. This study examined the construct of the OWB scale in the context of organizational study among 1745 Laotian employees. Second, it provides an extensive literature review on OWB, which has been retooled for developing scale validity. Third, a reliable OWB scale was constructed using EFA and CFA. Lastly, the valid construct of the six-dimensional model for OWB is essential to further the study of organizational employees.

## 2. Context and Theoretical Constructs

### 2.1. Context of Occupational Well-Being in Laos

Laos has the ninth-smallest workforce in South East Asia, with the greatest challenges for young employees in the socialist labor market. There are a total of 3.07 million (79.3%) people categorized as aged between 18 and 35 years in the workforce [34]. The majority of Laotian employees have a daily income of USD 9.50–11.50 and they are a low-skilled workforce. Moreover, adherence to low-skilled labor income has been found to be important for OWB in organizational employees [27,29]. The OWB construct is based on employee growth and fulfillment in three dimensions: job satisfaction (e.g., satisfied with the job), work engagement (e.g., personal judgment), and subjective well-being (e.g., complete physical, mental and social well-being).

Previous studies on OWB scales for Laotian employees have measured them through human well-being [26,35]. They have attempted to construct five dimensions: family well-being (e.g., my well-being is reliant on my work and family), community well-being (e.g., living in harmony with fairness and equality), health well-being (e.g., happiness both physically and mentally), natural well-being (e.g., living with safety from natural abandonment), and living well-being (e.g., having employment and no debts). Previous constructs measured have been OWB constructs on psychological well-being scales [36]. For instance, a common practice when measuring psychological well-being at work is to take OWB items (e.g., Have you been feeling on top of the work?).

As for the universalist–relativist construct, having job satisfaction can be measured as an experience of the OWB scale [35], particularly in socialist societies such as Laos (where a large proportion of the workforce are low-skilled) [34,37]. In fact, the term OWB is often measured (e.g., the ability to achieve work satisfaction or financial reward via work) [19]. OWB constructs have integrated a global indicator, which in the Laotian context is supported to meet the basic needs of employee well-being as a key resource of job satisfaction [38,39]. Together, the six dimensions represent the employee perspective of OWB. Therefore, to truly capture an organizational perspective regarding overall OWB at work, it is important to develop a work-specific measure of employee well-being.

### 2.2. Affective Well-Being

Previous research on the affective well-being of employees has been well constructed in organizational studies [6]. Originally, affective well-being was constructed in terms of both the positive and the negative well-being of employees [40]. Daniels [41] measured affective well-being based on positive affect (comfort, pleasure, calm, enthusiasm and vigor) and negative affect (anxiety, depression, boredom and tiredness). Boddy [42] constructed it in a binary way: anxiety–comfort, depression–pleasure, and tiredness–vigor. Furthermore, an excerpted construct of previous studies [43] has measured positive well-being (satisfied, relaxed, energetic, enthusiastic and inspired) and negative well-being (depressed, anxious, disgusted, frustrated and gloomy).

A large number of studies on affective well-being have been constructed in organizational studies [44,45,46,47]. For instance, Russell and Daniels [48] and Spieler et al. [49] developed general scales of positive well-being (e.g., the ability of employees to make choices that influence when, where and for how long they engage in delight, elation, relaxation and serenity) and negative well-being (e.g., right now, that is, at the present moment the person is down, annoyed, nervous and lethargic). Ribeiro et al. [50] and Sora et al. [51] attempted to measure psychological scales of positive emotions (e.g., enjoyment, interest and contentment) and negative emotions (e.g., identification with and involvement in emotional attachment).

### 2.3. Professional Well-Being

The professional well-being of employees in organizational studies has been constructed in the literature as aspiration, competence, autonomy, and working life [12]. Some scholars have constructed professional well-being regarding the notion of job-related motivation, ambition, achievement, and performance outcomes [14,24,52]. Meanwhile, professional well-being has been measured as work centrality, job performance, job satisfaction, successful job transitions, and financial standing. There are a number of well-established factors available to assess professional well-being (e.g., growth, purpose in life, autonomy, work productivity, and competence) [53,54].

To measure the construct comprehensively, professional well-being in organizational studies has frequently extracted common items (e.g., a creative organization really cares about motivation, achievement, work productivity, and performance) [52,55]. Meanwhile, the latter constructs the professional well-being scale of employees (e.g., I definitely want a career for myself in organizational work outcomes) [56]. For example, Eatough et al. [57] developed the main constructs of professional well-being based on self-esteem in organizing employees (e.g., at the moment, I feel that I have a number of good qualities). This type of measurement of professional well-being is a complex construct of work comfortability, work motivation, and work behavior [58].

### 2.4. Social Well-Being

Some scholars have measured the social well-being of employees while focusing on current research in organizational studies [26,33,59]. Various studies have constructed social well-being as an essential assessment of acceptance, coherence, contribution, actualization and integration [60]. Empirical research has constructed the social well-being of employees to measure access to jobs, labor participation, employment trust, social support, and networks [61,62,63]. In the current scenario, social well-being is a complex construct of social norms, social functions, and depersonalization.

From the literature review, a core construct of social well-being to measure organizational employees is the key mark of engagement (e.g., I engage with my colleagues, employers, and organization) and function (e.g., I feel comfortable in my interactions with my colleagues, I receive social support, and I am part of the organization) [7,64]. For instance, social well-being has been constructed in current studies [24,65] to measure actualization and integration (e.g., I love to spend time with my colleagues), interaction (e.g., employees are proud to tell people about the organization), and intensification (e.g., the extent to which employees feel that their job requires them to work with social organizations).

### 2.5. Cognitive Well-Being

Empirical research on the cognitive well-being of employees has been well constructed within the domain of organizational studies [6]. Most prior research on cognitive well-being has constructed the domain of work life, life circumstances, and work absorption [12,66,67]. Some scholars have constructed cognitive well-being while focusing on the experience of life satisfaction (e.g., expected continuity and performance of the job) and job satisfaction (e.g., perceived and desired continuity of the job). For instance, Warr [7] suggested a notion of cognitive well-being that measures positive reappraisal, a refocus on planning, job acceptance, and purpose in life of organizational employees.

Measuring cognitive well-being is constructed using work cognition, cognitive perception, and cognitive engagement [33,68,69]. To assess, the construct of employees substantially measures cognitive function (e.g., the degree to which employees are able to assimilate new information and to concentrate on their work) [12]. This helps to pave the way for research that clarifies how cognitive well-being is assessed (e.g., I can concentrate easily) [70]. By measuring various organizational employees’ cognitive well-being, social cognition has constructed self-efficacy (e.g., employees’ judgments of their capabilities to organize and execute courses of action required to attain designated types of performance) [71].

### 2.6. Psychological Well-Being

Research evidence suggests that the psychological well-being of employees plays a significant role in measuring organizational studies [33,50,72]. The measurement of psychological well-being has constructed the notions of work involvement, employee engagement, and job satisfaction [73]. Loon et al. [74] and López et al. [75] measured how the key constructs of psychological well-being are environmental mastery, life purpose, self-acceptance, and job autonomy. To date, many scholars have measured psychological well-being in different dimensions such as thriving at work [76], perceived organizational support [77], and organizational behavior [38].

Kundi et al. [78] measured three dimensions of organizational employees: hedonic well-being (e.g., my life conditions are excellent), eudaimonic well-being (e.g., my life is centered on a set of core beliefs that give meaning to my life), and job performance (e.g., I try to work as hard as possible). Zheng et al. [79] measured psychological well-being (e.g., I generally feel good about myself and I am confident). A number of scale constructs have been developed to assess various forms of psychological well-being (e.g., in general, I feel that I am content with the organization in which I am employed) [80].

### 2.7. Psychosomatic Well-Being

The construct of the psychosomatic well-being of employees has received widespread attention in organizational studies [17,33,81]. Prior research has measured how psychosomatic well-being tends to assess backache, headache, fatigue, and an upset stomach [82]. Consistent with prior work, Rahimnia and Sharifirad [83] have constructed job satisfaction, emotional exhaustion, and psychosomatic strain. Furthermore, the psychosomatic well-being of employees has assessed positive health complaints (e.g., feeling, cheerfulness, and coping ability), negative health complaints (e.g., headache, muscular pain, and cardiovascular functioning), and mental health complaints (e.g., mood, stress, and depression) [17,84,85].

Psychosomatic well-being assesses physical complaints (e.g., upset stomach and headache) and mental complaints (e.g., feeling tired and difficulty in concentrating) [86]. Some studies have measured psychosomatic complaint items (e.g., please tell me whether the following health complaints have occurred in the last 12 months while working and on working days) [87]. Franke [88] constructed psychosomatic health problems, which were measured as headache, fatigue, weariness, stomach and digestive complaints, tension and irritability, sleep disorders, dejection, physical exhaustion, and emotional exhaustion. Some items were developed by Pereira and Elfering’s [89] psychosomatic well-being scale (e.g., How would you rate your health complaints in the preceding 30 days?).

## 3. Methods

### 3.1. Data and Sample

This study was conducted among organizational employees in Vientiane, Laos. The data were collected via labor force surveys, comprising a random sample whose ages ranged from 18 to 60 years in the labor market [90]. The initial stage of data collection included 40,000 employees in five occupational categories (full-time and part-time employees, casual employees, fixed-term and contracted, apprentices and trainees, and commission and piece-rate employees). The second stage included 25,000 employees (62.5%) who constitute the potential labor force in Vientiane, Laos. To qualify for data collection, the individual had to be employed in the industrial and household sector, of which 10,000 employees are currently employed in the labor market. In the final stage, samples came from nine districts in Vientiane, according to the labor force survey, consisting of own-account employees, contribution family employees, and industrial employees.

Using systematic sampling, a total of 2000 employees were invited to participate in the structured interviews. The data collection consisted of a sample of 1745 volunteers who participated in the study. The refusal rate was approximately 12.75%, amounting to 255 subjects. Before the structured interviews commenced, there was a follow-up to examine the scale’s test reliability. The structured interviews were pretested with 30 participants. Demographic data from the samples were not gathered in the same way, and participants were informed that the schedules for the interviews were based on volunteers’ convenience. The demographic characteristics of the samples included in the study are presented in Table 1.

### 3.2. Constructing Scale Development

To construct scale validation, DeVellis [91] recommended three reasons: (i) several existing scales have not been produced using valid procedures, (ii) there is different research from the literature to apply in the context, and (iii) constructs have not yet been in an existing scale. Previous original studies have developed scales to assess the OWB scale, namely existing scales for psychological well-being [92], mental well-being [6] and affective well-being [7]. Later, Van Horn et al. [12] introduced a theoretical basis of OWB constructs in five dimensions: affective well-being, professional well-being, social well-being, cognitive well-being, and psychosomatic well-being.

Pradhan and Hati [25] developed employee well-being constructs (psychological, social, subjective and workplace) in organizational studies. In Laos, research to measure employees’ OWB continues to lag behind in comparison to needs on an organizational level. Previous studies have had to make trade-offs when constructing their studies, including 10-item OWB scales [27], a small sample of non-representative groups [3], and undefined procedures of validity [2]. Despite the shortcomings mentioned above, some scales used to measure OWB have not been new scales, such as that of Laotian employees. Doble and Santha’s [93] construction of the OWB scale is consistently assessed from the organizational perspective.

In sum, we identified eight stages to develop a rigorous scale: (i) the identification of a domain and item generation, (ii) content validity, (iii) pre-testing questions, (iv) item reduction, (v) the extraction of factors, (vi) a test of dimensionality, (vii) a test of reliability, and (viii) a test of validity [94]. Moreover, the study notes the importance of appropriate constructs and how one should be population- and context-relevant. There is also a need to develop and test validity which considers the multiple dimensions of OWB at organizational levels [4,23,24,93], hence the significant gap in the literature concerning OWB constructs, which provides appropriate tools for scale development. Table 2 depicts previous research on OWB dimensions and their various constructs.

### 3.3. Instruments

To ensure the construct of the OWB scale, three professional experts on OWB were selected for the structured interviews to evaluate “the index of item-objective congruence”. Expert rating accuracy was to evaluate the content validity of the items in the scale development stage. Three experts evaluated each item range as 1 for a clearly measuring objective, −1 for not a clearly measuring objective, or 0 for an unclear objective. According to Turner and Carlson [96], this study provides three stages for an item–objective congruence test: development stage (expert judgments of items), validation stage (selecting valid items), and validation evaluation (final instrumentation items).

The structured interviews were conducted while asking the participants to evaluate each item, the instructions, and the response scale for the instrument. Ericsson and Simon [95] suggested that structured interviews be conducted using think-aloud procedures as participants answer the questions. Specifically, the researchers read each question aloud to the participants and then recorded the processes that they implemented in arriving at an answer to a question. All structured interviews were conducted at the home addresses of the participants. The structured interviews took an average of 30–40 min and were conducted by the researcher and assistants as the participants finished.

### 3.4. Specify the Construct and Item Generation

This study closely follows a standard scale development [32] recommended to measure different constructs of an OWB scale [2,11,12,13,30]. Each of Churchill’s [32] seven recommended steps of scale development were implemented in this study, as can be seen in Table 3. As Churchill [32] recommended, the first step of scale development is to perform an extensive literature review. The construct of the scale has multiple dimensions [6,7] with a conceptual article on OWB (see Table 3). Based on previous work conducted by Churchill [32], a second step is recommended for generating a pool of items to construct item–objective congruence of the OWB scale in the pilot study.

Prior to conducting a pretest of the structured interviews, 30 samples were collected to gather feedback on questions, scales and clarity, mostly regarding the precision of items. Churchill [32] recommended the third and fourth steps for scale item generation using statistics of EFA. Utilizing systematic sampling, data from 1745 structured interviews were distributed door to door to residents residing in Vientiane, Laos, from 5 to 7 September. Churchill [32] recommended that the validity-related measures of 47-item OWB scales be assessed via an examination of the alpha coefficient [97]. To ensure scale validity, all items were rated on a five-point Likert-type scale ranging from 1 (strongly disagree) to 5 (strongly agree). The internal consistency reliability of the OWB scale was satisfactory, which produced the coefficient alpha (α = 0.70–0.91).

### 3.5. Analyses

The construct of the OWB scale was tested using CFA, which was carried over from the EFA using Stata 17 [98]. To establish the validity of the OWB scale, the descriptive statistics of the data were tested. Second, the initial model was tested for each factor solution with a data fit model. To modify the model fit, several goodness-of-fit indices of the CFA model were tested: the goodness-of-fit index (GFI), the Tucker–Lewis index (TLI > 0.95), the comparative fit index (CFI > 0.95), the reported root mean square residuals (RMSR < 0.05), the root mean square error of approximation (RMSEA < 0.05), Akaike’s information criterion (AIC), the Bayesian information criterion (BIC), the ratio of chi-square (χ^2^) to degrees of freedom, and the coefficient of determination ($R^{2}$) [99].

## 4. Results

### 4.1. Preliminary Analysis

Initially, this study was conducted to measure six-dimensional constructs of employees’ OWB scale. All of these constructs are aligned with the model, which maps to the same factor as a priori to produce an accurate reflection of the samples. The number of constructs, expert rating accuracy, mean, standard deviation, and Cronbach’s alpha are depicted in Table 4. The interrelation among the six dimensions showed that affective well-being was significantly related to professional well-being (*r* = 0.38, *p* < 0.01), social well-being (*r* = 0.31, *p* < 0.01), cognitive well-being (*r* = 0.35, *p* < 0.01), psychological well-being (*r* = 0.68, *p* < 0.01) and psychosomatic well-being (*r* = 0.59, *p* < 0.01), as can be seen in Table 5.

### 4.2. Exploratory Factor Analysis

#### 4.2.1. EFA Model Development

The six dimensions of OWB constructs were subjected to an EFA model using Stata 19. An iterative process was undertaken to achieve the best factorial structure. The initial EFA resulted in a Kaiser–Meyer–Olkin (KMO) test through which to measure the six-dimensional OWB constructs (*a* = 0.91), with Bartlett’s test (χ^2^ = 862, *p* < 0.01) explaining 64.58% of the total variance. As they were generated in a large pool of items, the strength of the items was statistically tested to test cross-validity among the samples. The EFA iteration of the KMO measure was conducted with the 47 remaining items, which were loaded onto the six dimensions of the OWB construct.

The results of EFA to test the six-dimensional OWB constructs were aligned with the theoretical model. As expected, the EFA model enabled discerning a six-factor structure among the 14 variables, with the scale consisting of OWB-47. The KMO coefficients for the constructs of affective well-being, professional well-being, social well-being, cognitive well-being, psychological well-being, and psychosomatic well-being are 0.85, 0.79, 0.82, 0.86, 0.80, and 0.77, respectively, with each scale’s Bartlett test being significant at the 0.05 level. These EFA tests indicate adequate correlations among the items to be scaled, which are appropriate to perform using a CFA model.

#### 4.2.2. Cross-Validity of EFA

Cross-validity analysis was conducted to test whether the EFA prevailed in different samples. This EFA was tested through a cross-analysis of two samples: male (*n* = 790) and female (*n* = 955). The first group ranged in age from 18 to 35 years, while the second group ranged from 36 to 60 years. All samples (gender versus age) were found to reasonably fit with the dataset for sampling adequacy (KMO = 0.901) and sphericity (χ^2^ = 1929, *p* < 0.01) in explaining 60.4% of the total variance. The cross-validity between gender and age is presented in Table 6.

#### 4.2.3. Purified Construct of Item Generation

The purified construct of item generation was conducted to reveal the scale, item–total correlations, and the reliability of the six-dimensional OWB constructs as applied to the EFA model. The EFA (primarily principal component extraction and varimax rotation) was conducted to assess the internal consistency of the scale. The results of the Bartlett test (χ^2^ = 679.95, *p* < 0.01) provided solid justification for the EFA model. The reliability of the KMO measure for sampling adequacy is 0.92, explaining 67.51% of the six-dimensional OWB constructs. The findings of the EFA representing the respective factor loading of 47 items along with the OWB scale are presented in Table 7.

### 4.3. Confirmatory Factor Analysis

#### 4.3.1. CFA Model Development

The development of OWB scales has six-dimensional constructs, namely affective well-being, professional well-being, social well-being, cognitive well-being, psychological well-being, and psychosomatic well-being. The different scales of unidimensional OWB constructs constitute a relatively poor model (χ^2^ = 112.04, df = 7, *p* > 0.26, CFI = 0.75, TLI = 0.74, AIC = 126.505, BIC = 136.19, SRMR = 0.45, RMSEA = 0.49, and 95% CI [−0.10, 0.15]). Presented are the correlated scales between F1 and F2 (*β* = 0.40, *t* = 9.65, *p* < 0.01), F3 (β = 0.50, *t* = 11.04, *p* < 0.01), F4 (*β* = 0.52, *t* = 11.66, *p* < 0.01), F5 (*β* = 0.40, *t* = 7.72, *p* < 0.01) and F6 (*β* = 0.70, *t* = 18.39, *p* < 0.01). Table 8 shows the six-factor structure of different OWB scales. The initial standardized CFA for developing unidimensional constructs can be seen in Figure 1.

#### 4.3.2. Initial CFA Model

Prior to developing the initial model, the CFA was inspected to ensure that the model had converged to a proper solution. All constructs were well initiated, as it failed to achieve significance (*p* > 0.05), remaining below the 0.30 criterion cut-off. In the traditional CFA, the unidimensional model was not able to provide acceptance (Δχ^2^ = 219.09, Δ*df* = 25, ΔCFI = 0.92, ΔRMSEA = 0.40). To put the results in six-dimensional constructs, the model is appropriate for the data (χ^2^ = 312.19, *df* = 29, *p* > 0.01, CFI = 0.97, TLI = 0.97, AIC = 334.44, BIC = 439.22, SRMR = 0.03, RMSEA = 0.02, and 95% CI [2.73, 4.01]). Table 9 depicts a comparison between the models.

This difference in the goodness-of-fit model calls into question the appropriateness of the CFA for developing a fit model. Of particular relevance is that the factor correlation is substantially smaller in the initial model solution—compared with the fitness model solution—and more consistent with theoretical objectives. Modification indices (MI) showed that the model fit would improve standardized factor loadings within a range (*β* = −1.25–1.26). F1 and F2 showed a poor model fit with existing data of χ^2^ values (MI = 153), between the constructs of F3 and F4 (MI = 269) and between the constructs of F5 and F6 (MI = 199). Figure 2 depicts the initial CFA for the OWB model.

#### 4.3.3. Goodness-of-Fit Model

The six-dimensional OWB constructs identified in the model were further confirmed by means of CFA through Stata 19. All items in the OWB scale were significantly loaded onto their respective construction of the CFA model. The modification indices that the data showed (χ^2^ = 502.26, *df* = 31, *p* > 0.01, CFI = 0.98, TLI = 0.98, AIC = 416.36, BIC = 418.69, SRMR = 0.02, RMSEA = 0.01, and 95% CI [3.04, 5.19]) achieved the model threshold. All constructs of OWB scales had acceptable reliability with composite reliability above 0.65 and a coefficient of determination above 0.5. The final CFA illustrated a reasonable-fit model, although as compared to the six-dimensional OWB constructs, 14 variables and 47 items produced an acceptable model, as can be seen in Table 10. The final CFA model of modification indices is shown in Table 11.

The first construct of F1 indicated a poor fitness of model fit indices (χ^2^ = 45.71, *df* = 13, *p* > 0.05, RMSEA = 0.19, 95% CI [−0.350, 0.01]). The second construct of F2 illustrated a low model fit index (χ^2^ = 94.71, *df* = 18, *p* > 0.05, RMSEA = 0.15, 95% CI [−0.010, 0.28]). The third construct of F3 indicated a slightly acceptable-fit model (χ^2^ = 120.90, *df* = 21, *p* > 0.05, RMSEA = 0.09, 95% CI [0.05, 0.26]). The fourth construct of F4 contained an adequate-fit model (χ^2^ = 189.14, *df* = 23, *p* > 0.05, RMSEA = 0.08, 95% CI [0.23, 0.51]). The fifth construct of F5 observed a fit model (χ^2^ = 272.95, *df* = 25, *p* > 0.01, RMSEA = 0.05, 95% CI [0.67, 0.97]). The final construct of F6 offered an excellent-fit model (χ^2^ = 272.019, *df* = 25, *p* > 0.01, RMSEA = 0.04, 95% CI [0.81, 1.24]).

The correlated values of the six-dimensional OWB constructs are presented in Table 12. The full CFA model of the six-dimensional OWB constructs is depicted in Figure 3. The full six-dimensional model shows the goodness-of-fit data (χ^2^ = 357.03, *df* = 30, *p* > 0.01, CFI = 0.97, TLI = 0.98, AIC = 434.22, BIC = 441.92, SRMR = 0.04, RMSEA = 0.02, 95% CI [2.79, 4.56]), which meet the values of required fit indices. Presented is the final construct of F1 ($R^{2}$ = 0.71, *p* < 0.01), F2 ($R^{2}$ = 0.78, *p* < 0.01), F3 ($R^{2}$ = 0.85, *p* < 0.01), F4 ($R^{2}$ = 0.79, *p* < 0.01), F5 ($R^{2}$ = 0.91, *p* < 0.01) and F6 ($R^{2}$ = 0.72, *p* < 0.01). The CFA thus confirms that the OWB scale has 47 items with 14 variables of six-dimensional constructs.

## 5. Discussion

### 5.1. Discussion of Key Results

This study provides a scale development process for OWB constructs, with a focus on using a CFA model. The constructs of scale validation were tested with Laotian samples (own-account employees, contribution family employees, and industrial employees). The study examined the OWB constructs comprising the six dimensions with 14 variables of OWB-47 in Laos. Specifically, the study assessed the scale validity as well as reliability of the OWB scales. The CFA model showed support for a stable multidimensional OWB scale which supports developing an initial model. Overall, the findings showed that the OWB model is a robust goodness-of-fit model (configurable, metric, scalar, and error invariance).

The valid construct of OWB scales comprises affective, professional, social, cognitive, psychological and psychosomatic well-being [6,7,11,12,13,15]. Recent research has underscored the complexity of the construct of OWB, testing the CFA model [2,31]. Consequently, the important aspects of this construct are dispersed across a variety of instruments. We found six-dimensional OWB using well-established measures of model comparison. The OWB model has acceptable constructs: high positive correlations with the employee well-being scale [23,24], psychological well-being [6,7] and eudemonic well-being [84].

Previous studies have developed international OWB scales [99,100] but without comparing model measures. Thus, this study carried out testing using EFA to confirm with CFA the theoretical structure of OWB constructs, thereby developing a valid scale for research practice. As in previous studies [12,27,44,62,100] which have measured OWB constructs of organizational employees, the correlations among the six-dimensional model were all statistically significant. Some studies have constructed the validity of occupational characteristics [2], psychological well-being items [16], psychosomatic well-being [17] and employee well-being [24]. Our valid six-dimensional model began with a coherent theory which led to the development of valid psychosomatic instruments.

Clearly, there are many useful and valid measures of OWB available to researchers and academicians. The findings from this CFA model, however, support that the development process—a six-dimensional OWB scale—is a reliable and valid measure for the assessment of employee well-being with good psychometric properties. In addition, the psychometric validity of Ryff’s psychological well-being items [16,17,19,23] correlated meaningfully with a modified version, which could then be seen as non-redundant validation of the new measure. That is, the measure (which is successful in addressing the needs of the scale) is good, with well-distributed scores robust to demographic variables. The broad scope of the new measure means that it has considerable scope for use in both research and organizational settings.

### 5.2. Theoretical Implications

This study has various theoretical implications which offer important insights into measuring OWB scales. The key findings have filled the theoretical gap in OWB constructs [6,7,12,23,47,67]. Most studies have constructed an assessment of OWB-10 items, affective (12-item), professional (7-item), social (7-item), cognitive (7-item) and psychosomatic (23-item), to measure well-being scales [12,27,100]. Previous scholars have theoretically constructed the job-related affective well-being scale [7], psychological well-being scale [3,4,101], and OWB index [102].

A key theoretical contribution of this study is that of measuring an OWB scale [18,38] in the broader context of organizational employees. Most studies have essentially measured general well-being measures [64], subjective well-being indicators [103,104] and the workplace well-being scale [24]. However, there has been no evidence regarding context-free constructs of the OWB scale [23]. Thus, the authors aligned well-being theory [10] with the remeasured six-dimensional OWB constructs [6,7,12]. We theoretically measured the OWB constructs [12,23] in the context of Laotian employees [27].

### 5.3. Managerial Contributions

This study makes some key managerial contributions. First, on the individual level, this construct can serve as a psychometric tool that enables organizational employees to measure their OWB. Not only does it improve organizational employees, but it also provides a solid foundation for developing their OWB. More precisely, this study shows that employees are an integrative construct of OWB-47, which tests their use of the approach put forward by Churchill [32]. The model offers a new angle—and a psychometric measurement tool—for organizational employees to use when implementing OWB initiatives. The model shows that the OWB construct comprises affective, professional, social, cognitive, psychological and psychosomatic well-being, implying that organizational employees should not only focus on life satisfaction, but also show genuine concern for well-being.

Second, on the managerial level, managers can use the scale to understand employees’ level of OWB and intervene in a timely manner. Such efforts may help organizations to avoid a tragic situation, such as employees’ affective (positive and negative) well-being, to promote more job autonomy, job satisfaction, and life domains, whereby reducing psychosomatic complaints and symptoms. Lastly, on the organizational level, for employers that implement psychometric programs for OWB practices, it may enhance their long-term levels of organizational employee effectiveness.

### 5.4. Limitations and Future Research

It is worth mentioning the limitations of the current study. First of all, this study was performed using a CFA model of six-dimensional OWB constructs; thus, there needs to be a rigorously developed and validated scale instrument for the general population. Furthermore, the scale was validated in only three groups (own-account employees, contribution family employees, and industrial employees); therefore, it needs to be measured in different occupations. Future research could explore how different groups impact on OWB amongst larger organizational employees, where employee well-being may be less well defined and, thus, the role of managers could be even more significant.

Second, the samples were collected in Vientiane, which is not sufficient to generalize the findings across Laos. Moreover, the current samples consisted of mostly females (66.19%). The relevance of the OWB scale in the context of males in particular would be worth exploring. Thus, future research may implement the study amongst all groups that are represented throughout the nation, along with a demographically diverse sample of organizational settings. Third, the valid construct of OWB scales comprising the six dimensions with 14 variables of OWB-47 items needs to be further investigated in different cultures. Future research should verify the OWB constructs by retesting them as linear hypotheses with longitudinal designs, whose findings can be a theorized valid scale with the existing empirical model.

## 6. Conclusions

The purpose of this study is to develop a scale validation of OWB constructs with Laotian samples. The findings show that the OWB scale is a useful instrument with which to adequately measure the context of Laotian employees. The scale validation of OWB constructs is developed and validated for measuring affective well-being, professional well-being, social well-being, cognitive well-being, psychological well-being and psychosomatic well-being. The study establishes the importance of OWB-47 by means of a CFA model, and the construct clarifies its relationship with theoretically relevant models, which significantly relates to the study samples. In sum, these findings serve as a strong CFA model in which the validity of OWB-47 can be extended in order to assess Laotian own-account employees, contribution family employees, and industrial employees.

## Figures and Tables

**Figure 1 behavsci-14-00248-f001:**
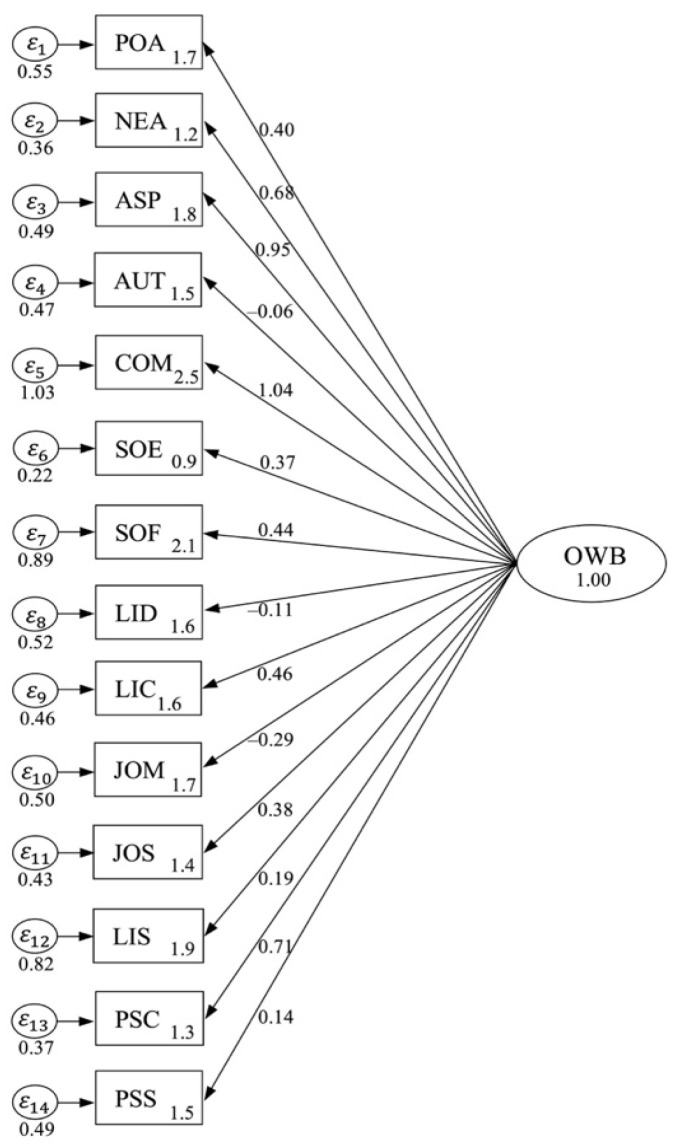
A unidimensional CFA for OWB model.

**Figure 2 behavsci-14-00248-f002:**
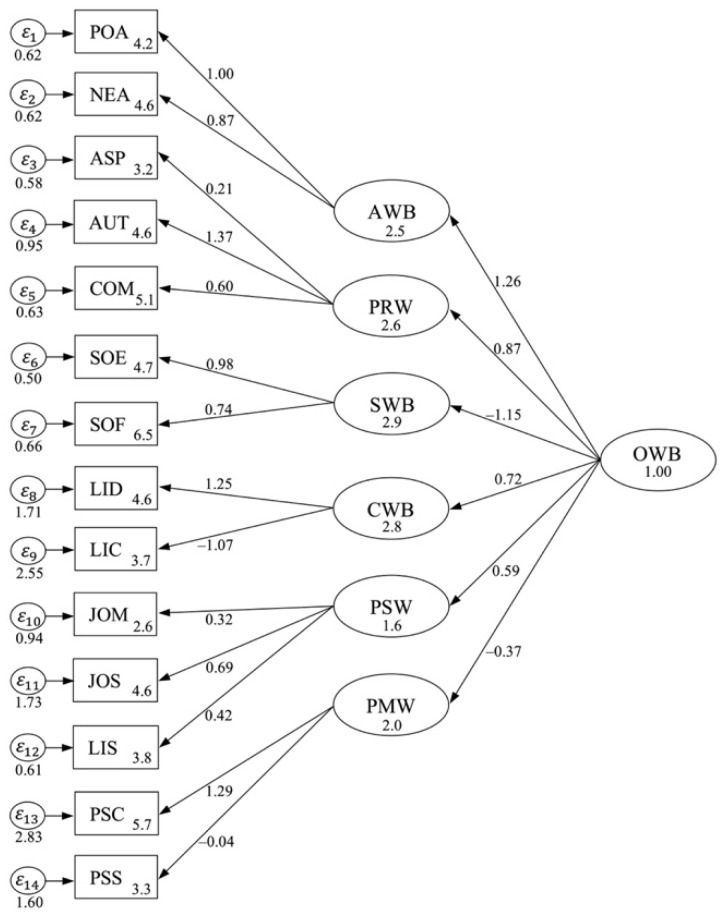
Initial CFA for OWB model.

**Figure 3 behavsci-14-00248-f003:**
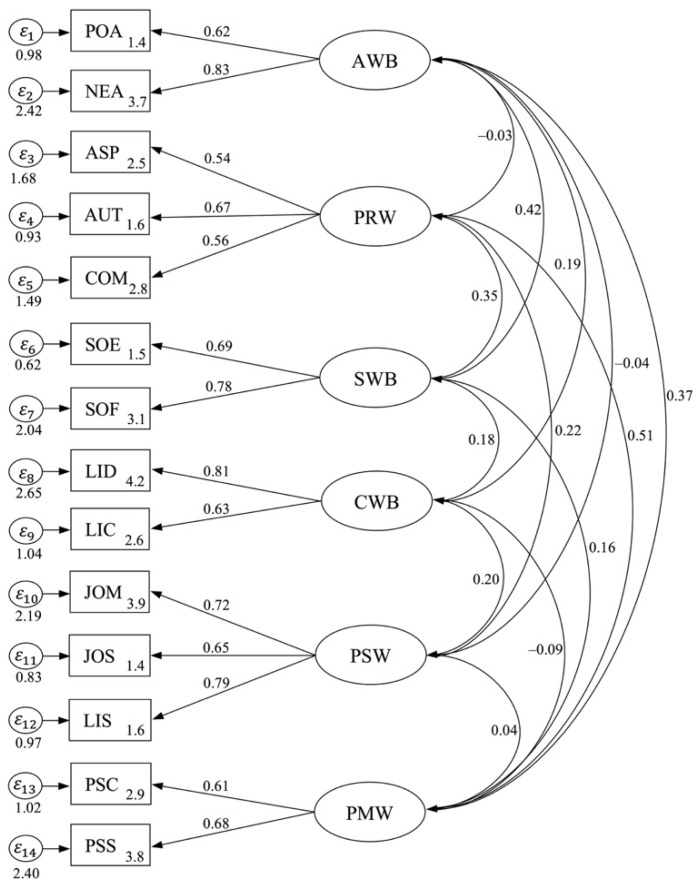
CFA for OWB model.

**Table 1 behavsci-14-00248-t001:** Demographic samples.

Characteristics	Demographic Categories	Employees (%)
Manufacturing		550 (31.5)
Family sector		518 (29.6)
Wholesale and retail trade		290 (16.6)
Services		205 (11.7)
Construction		102 (5.8)
Transportation and storage		80 (4.5)
Gender	Male	590 (33.81)
	Female	1155 (66.19)
Age (years)	18–25	685 (39.2)
	26–35	912 (52.2)
	36–45	98 (5.6)
	46–60	50 (2.8)
Marital status	Married	1305 (74.7)
	Single	440 (25.3)
Educational level	Primary school	409 (23.4)
	High school	679 (38.9)
	Vocational training school	657 (37.6)
	Bachelor’s degree	149 (8.5)
Employment status	Full-time	901 (51.6)
	Part-time	252 (14.4)
	Shift workers	90 (5.1)
	Daily hire and weekly hire	397 (22.7)
	Outworkers	105 (6.1)
Working hours per week	15–25	121 (6.9)
	26–36	201 (11.5)
	37–48	919 (52.6)
	>49	504 (28.8)
Monthly income	<LAK 1,500,000	200 (11.4)
	LAK 2,000,000–3,000,000	819 (46.9)
	LAK 4,000,000–5,000,000	496 (28.4)
	>LAK 5,000,000	230 (13.1)

USD 1: LAK 8500 (2019).

**Table 2 behavsci-14-00248-t002:** Previous research on developing OWS constructs.

Authors	OWB Dimensions	Scale Constructs
Warr [6,7], Van Horn et al. [12], Daniels [41], Basińska et al. [43], Boddy [42], Van Katwyk et al. [43]	Affective well-being	1. Positive affect (POA) 2. Negative affect (NEA)
Van Horn et al. [12], Novaes et al. [95]	Professional well-being	1. Aspiration (ASP) 2. Autonomy (AUT) 3. Competence (COM)
Van Horn et al. [12], Boreham et al. [63]	Social well-being	1. Social engagement (SOE) 2. Social function (SOF)
Van Horn et al. [12], Kuykendall et al. [71]	Cognitive well-being	1. Life domains (LIDs) 2. Life circumstances (LICs)
Bretones and Gonzalez [3], Khatri and Gupta [23], Joo et al. [67]	Psychological well-being	1. Job motivation (JOM) 2. Job satisfaction (JOS) 3. Life satisfaction (LIS)
Van Horn et al. [12], Åslund et al. [17], Pereira and Elfering [89]	Psychosomatic well-being	1. Psychosomatic complaints (PSCs) 2. Psychosomatic symptoms (PSSs)

**Table 3 behavsci-14-00248-t003:** Scale development procedures.

Stage	Recommended Procedure	Technique Implemented
1	Specify domain of construct	Theoretical scale construction on occupational well-being
2	Generate pool of items	Literature search
		Item–objective congruence for testing the pilots
		Structured interview pretest (*n* = 30)
3	Collect data	Structured interview test in Vientiane, Laos (*n* = 1745)
4	Purify measure	Exploratory factor analysis used for factorial structure of the scale—identified four major dimensions of OWB with 47 items
5	Assess validity	Cross-validity analysis, factorial invariance, and model development (albeit on different scales)
6	Assess reliability	Confirmatory factor analysis (construct reliability: initial model, unidimensional model, and goodness-of-fit model)
7	Develop norms	Model practical implications

Source: scale development procedures adopted from Churchill [32].

**Table 4 behavsci-14-00248-t004:** Descriptive analysis.

OWB Construct	No. of Items	Mean Expert Rating Accuracy	Calibration Sample (*n* = 1000)			Validation Sample (*n* = 1745)		
			Mean	SD	*a*	Mean	SD	*a*
Affective well-being	12	4.7	2.98	0.93	0.79	3.20	0.85	0.82
Professional well-being	9	4.5	3.75	0.87	0.82	3.95	0.72	0.85
Social well-being	6	4.6	3.51	0.83	0.80	3.67	0.87	0.83
Cognitive well-being	6	4.4	3.02	0.71	0.74	3.79	0.80	0.70
Psychological well-being	9	4.7	4.10	0.78	0.90	4.23	0.77	0.91
Psychosomatic well-being	5	4.5	3.46	0.84	0.71	3.58	0.80	0.89

**Table 5 behavsci-14-00248-t005:** Intercorrelation matrix.

OWB Construct	F1	F2	F3	F4	F5	F6
F1. Affective well-being	1.00					
F2. Professional well-being	0.38 *	1.00				
F3. Social well-being	0.46 **	0.31 *	1.00			
F4. Cognitive well-being	0.31 *	0.49 **	0.35 *	1.00		
F5. Psychological well-being	0.45 **	0.67 **	0.51 **	0.68 *	1.00	
F6. Psychosomatic well-being	0.57 **	0.34 *	0.37 *	0.48 **	0.59 **	1.00

* *p* < 0.05; ** *p* < 0.01.

**Table 6 behavsci-14-00248-t006:** Cross-validity analysis (gender versus age).

Sample	Mean	SD	SE	CFI	NFI	IFI	RMSEA (95% CI)
Gender							
Male	0.75	1.04	0.03	0.89	0.88	0.90	0.55 (0.056, 0.065)
Female	1.02	1.78	0.04	0.91	0.92	0.92	0.55 (0.054, 0.062)
Age							
18 to 35 years	1.06	1.39	0.05	0.93	0.94	0.94	0.62 (0.056, 0.069)
36 to 60 years	0.89	1.27	0.03	0.94	0.95	0.96	0.62 (0.057, 0.070)

**Table 7 behavsci-14-00248-t007:** Final factorial structure of OWB scale.

Items	SFL	SRV
**Affective well-being**		
Positive affect		
OWB1: My job makes me feel satisfied	0.79	0.24
OWB2: My job makes me feel proud	0.85	0.68
OWB3: My job makes me feel energetic	0.75	0.36
OWB4: My job makes me feel enthusiastic	0.74	0.35
OWB5: My job makes me feel inspired	0.85	0.37
OWB6: My job makes me feel happy	0.78	0.29
Negative affect		
OWB7: My job makes me feel angry	0.88	0.31
OWB8: My job makes me feel depressed	0.89	0.38
OWB9: My job makes me feel anxious	0.87	0.44
OWB10: My job makes me feel disgusted	0.79	0.30
OWB11: My job makes me feel frustrated	0.89	0.33
OWB12: My job makes me feel gloomy	0.91	0.52
**Professional well-being**		
Aspiration		
OWB13: In my work, I seek new challenges	0.89	0.51
OWB14: To advance my job	0.84	0.48
OWB15: To be seen to be successful in the job	0.86	0.45
Autonomy		
OWB16: I make my own decisions at work	0.89	0.54
OWB17: I decide what I will do at work	0.87	0.47
OWB18: To decide my own priorities at work	0.89	0.54
Competence		
OWB19: I need considerable growth in my job	0.74	0.26
OWB20: I feel this is one of my strongest interpersonal skills and performances	0.82	0.38
OWB21: To work with interprofessional collaboration, colleagues, and teamwork	0.88	0.40
**Social well-being**		
Social engagement		
OWB22: I engaged with my colleagues	0.89	0.54
OWB23: I engaged with my employers	0.86	0.52
OWB24: I engaged with my organization	0.82	0.57
Social function		
OWB25: I feel comfortable in my interactions with workers and employers	0.82	0.57
OWB26: My colleagues ask me for advice and support	0.78	0.41
OWB27: I am an important part of my job and organization	0.81	0.46
**Cognitive well-being**		
Life domains		
OWB28: I am satisfied with my family life	0.78	0.33
OWB29: I am satisfied with my social life	0.77	0.42
OWB30: I am satisfied with my job and organization	0.79	0.35
Life circumstances		
OWB31: I have goals and ambitions	0.78	0.36
OWB32: Average for people of my age	0.79	0.58
OWB33: The best moments of my life are in the past	0.81	0.54
**Psychological well-being**		
Job motivation		
OWB34: I feel a sense of personal satisfaction when doing this job well	0.88	0.55
OWB35: I take pride in doing my job as well as I can	0.78	0.52
OWB36: I like to look back on the day’s work with a sense of a job well done	0.79	0.46
Job satisfaction		
OWB37: The freedom to choose my own method of working	0.81	0.44
OWB38: My chance of promotion	0.84	0.52
OWB39: My job security	0.86	0.46
Life satisfaction		
OWB40: I am satisfied with my life	0.80	0.42
OWB41: The conditions of my life are excellent	0.81	0.43
OWB42: In most ways, my life is close to my ideal	0.83	0.45
**Psychosomatic well-being**		
Psychosomatic complaints		
OWB43: I suffer from headaches	0.89	0.57
OWB44: I suffer from digestive trouble	0.75	0.54
OWB45: I suffer from dizziness	0.77	0.42
Psychosomatic symptoms		
OWB46: I have symptoms of poor mental health	0.79	0.46
OWB47: I have symptoms of poor physical health	0.76	0.38

OWB: occupational well-being; SFL: standardized factor loading; SRV: standardized residual variance; a: Cronbach’s alpha.

**Table 8 behavsci-14-00248-t008:** Factor predictions for the correlated scales, albeit on different scales.

OWB Constructs	F1	F2	F3	F4	F5	F6
F1. Affective well-being	1.00 (0.57)					
F2. Professional well-being	0.40 * (0.36)	1.00 (0.60)				
F3. Social well-being	0.61 ** (0.37)	0.50 ** (0.31)	1.00 (0.53)			
F4. Cognitive well-being	0.50 ** (0.78)	0.59 ** (0.56)	0.52 ** (0.40)	1.00 (0.67)		
F5. Psychological well-being	0.49 * (0.46)	0.46 * (0.76)	0.30 * (0.30)	0.40 * (0.45)	1.00 (0.88)	
F6. Psychosomatic well-being	0.72 ** (0.29)	0.51 ** (0.40)	0.44 * (0.43)	0.69 ** (0.34)	0.70 ** (0.42)	1.00 (0.57)

* *p* < 0.05; ** *p* < 0.01.

**Table 9 behavsci-14-00248-t009:** Comparison between CFA models.

	Goodness-of-Fit Indices								Versus	Factor Comparison Model			
	χ^2^	*df*	TLI	CFI	AIC	BIC	RMSR	RMSEA		Δχ^2^	Δ*df*	ΔCFI	ΔRMSEA
F1	173.29	26	0.95	0.95	354.21	369.35	0.04	0.05		–	–	–	–
F2	107.33	25	0.96	0.97	245.02	256.99	0.04	0.05	F2 vs. F1	189.84	25	0.97	0.05
F3	259.10	27	0.97	0.96	380.29	391.99	0.02	0.03	F3 vs. F1	271.99	26	0.97	0.03
F4	185.92	26	0.95	0.95	401.11	418.04	0.04	0.05	F4 vs. F1	216.44	25	0.96	0.03
F5	150.11	24	0.98	0.97	325.99	335.78	0.01	0.03	F5 vs. F1	189.46	24	0.98	0.00
F6	201.67	28	0.96	0.98	286.94	289.78	0.05	0.04	F6 vs. F1	211.47	25	0.97	0.04

**Table 10 behavsci-14-00248-t010:** Summary of goodness of fit for invariance tests.

Model Description	Comparison	χ^2^	Δχ^2^	*df*	Δ*df*	Sig.	CFI	ΔCFI	NNFI	ΔCFI ≤ 0.01	RMSEA (95% CI)
Initial variance	F3 vs. F2	5642.45	–	1246	–	–	0.905	–	0.901	–	0.56 (0.053–0.057)
Metric invariance	F4 vs. F3	5719.03	–	1393	–	–	0.908	<0.01	–	Yes	0.52 (0.052–0.055)
Scalar invariance	F5 vs. F4	–	40.46	–	36	*p* < 0.01	0.911	–	0.906	Yes	0.55 (0.051–0.054)
Error invariance	F6 vs. F5	6492.56	120.56	1450	44	*p* < 0.01	0.903	0.04	0.894	Yes	0.55 (0.052–0.056)
Fit variance											
Confiture invariance	F3 vs. F2	–	45.95	–	22	–	0.915	–	0.906	–	0.54 (0.053–0.054)
Metric invariance	F4 vs. F3	560.20	42.16	–	26	–	0.920	–	0.908	–	0.55 (0.054–0.056)
Scalar invariance	F5 vs. F4	–	41.45	1510	28	*p* < 0.01	0.910	<0.01	0.910	Yes	0.52 (0.051–0.053)
Error invariance	F6 vs. F5	5772.36	89.57	–	42	*p* < 0.01	0.950	0.01	0.920	Yes	0.53 (0.052–0.055)

**Table 11 behavsci-14-00248-t011:** CFA modification indices.

OWB Construct	*β* (SE)	z	P > |z|	95% CI
Means				
F1	4.56 ** (0.06)	73.50	0.00	[2.94, 4.29]
F2	6.39 ** (0.06)	89.99	0.00	[2.74, 4.05]
F3	3.46 ** (0.06)	36.45	0.00	[3.30, 4.93]
F4	2.42 ** (0.05)	45.75	0.00	[4.94, 4.29]
F5	4.35 ** (0.07)	58.66	0.00	[3.94, 4.85]
F6	4.57 ** (0.07)	66.44	0.00	[4.00, 5.24]
Loading				
F1	0.55 ** (0.10)	2.40	0.02	[0.45, 0.74]
F2	0.75 ** (0.11)	6.50	0.00	[0.49, 0.95]
F3	1.46 ** (0.07)	17.07	0.00	[0.87, 1.56]
F4	0.92 ** (0.05)	15.04	0.00	[0.82, 1.25]
F5	1.11 ** (0.15)	7.03	0.00	[0.87, 1.44]
F6	1.95 ** (0.19)	8.89	0.00	[0.69, 1.45]
Factor covariance				
F1	0.90 ** (0.16)	4.40	0.00	[0.61, 1.12]
F2	0.97 ** (0.12)	6.73	0.00	[0.85, 1.19]
F3	0.60 ** (0.08)	5.11	0.00	[0.22, 0.66]
F4	0.68 ** (0.07)	5.19	0.00	[0.21, 0.56]
F5	0.47 ** (0.03)	2.56	0.00	[0.06, 0.26]
F6	0.36 ** (0.05)	4.67	0.00	[0.15, 0.47]

** *p* < 0.01.

**Table 12 behavsci-14-00248-t012:** The correlated values of OWB constructs.

OWB Construct	F1	F2	F3	F4	F5	F6	% Endorsed
	*β* (SE)	*β* (SE)	*β* (SE)	*β* (SE)	*β* (SE)	*β* (SE)	
F1. Affective well-being							
POA	0.62 ** (0.07)						9.9
NEA	0.83 ** (0.08)						5.8
F2. Professional well-being							
ASP		0.54 ** (0.06)					13.56
AUT		0.67 ** (0.04)					15.77
COM		0.56 ** (0.05)					18.05
F3. Social well-being							
SOE			0.69 ** (0.06)				10.27
SOF			0.78 ** (0.08)				16.93
F4. Cognitive well-being							
LIDs				0.81 ** (0.09)			27.84
LICs				0.63 ** (0.07)			30.16
F5. Psychological well-being							
JOM					0.72 ** (0.05)		20.56
JOS					0.65 ** (0.09)		30.03
LIS					0.79 ** (0.06)		36.09
F6. Psychosomatic well-being							
PSCs						0.61 ** (0.09)	9.34
PSSs						0.68 ** (0.06)	7.80
Eigenvalue	1.66	1.60	1.53	1.48	1.57	1.32	Total
% Variance	9.54	9.43	8.75	7.12	8.10	7.83	50.77

** *p* < 0.01.

## Data Availability

Data are contained within the article.

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
