# Peer review of "Measuring Occupational Well-Being Indicators: Scale Construction and Validation"

_behavsci, 2024, doi:10.3390/bs14030248_

Round 1

Reviewer 1 Report

Comments and Suggestions for Authors

I found interesting the practical and theoretical assumptions that prompted the authors to undertake the study reported in this paper. All in all, I think the instrument constructed by the authors may prove useful for an exploratory survey of some of the underlying dimensions of the construct of occupational well-being, although I suspect it would be a rather cursory survey, given the very simple wording of the items.

I have only one note that I cannot help but report (premising that I have no personal or professional connection to the authors I am about to name nor to their instrument). Regarding the supposed absence of instruments assessing the psychological, psychosomatic, social, cognitive and organizational dimensions of occupational well-being, I would like to bring to the authors' attention that van Veldhoven and Meijman created in 1994 a very comprehensive and multidimensional questionnaire for the assessment of occupational well-being, the Questionnaire on the Experience and Evaluation of Work, then revised in 2015 and improved through the addition of other scales. The existence of this instrument clearly does not detract from the authors' work, but I think it would be appropriate to at least make mention of it in the literature review, especially in light of the statement in lines 42-43.

Comments on the Quality of English Language

I found it quite difficult to understand well what the authors wrote due to the construction of the sentences in English; I therefore recommend a revision of the English for greater clarity.

Author Response

Response to Reviewer 1

Dear Review,

Thank you for reviewing the manuscript entitled “Measuring Occupational Well-Being Indicators: Scale Construction and Validation”. We appreciate your suggestions. Following the comments, we have revised all suggestions from reviewers’ comments. Before re-submitting to the journal, we made a proofreading on this manuscript. All the comments are responded in the following parts.

Sincerely,

The author

==========================================

REVIEWERS’ SUGGESTIONS FOR THE AUTHOR:

Response to Reviewer 1

General comment:

I found interesting the practical and theoretical assumptions that prompted the authors to undertake the study reported in this paper. All in all, I think the instrument constructed by the authors may prove useful for an exploratory survey of some of the underlying dimensions of the construct of occupational well-being, although I suspect it would be a rather cursory survey, given the very simple wording of the items.

General response:

Thank you for taking the time to carefully review our manuscript. Having this feedback has helped us enormously to improve the entire manuscript. We believe the manuscript is now stronger and clearer as the reviewer suggested below.

Point 1:

I have only one note that I cannot help but report (premising that I have no personal or professional connection to the authors I am about to name nor to their instrument). Regarding the supposed absence of instruments assessing the psychological, psychosomatic, social, cognitive and organizational dimensions of occupational well-being, I would like to bring to the authors' attention that van Veldhoven and Meijman created in 1994 a very comprehensive and multidimensional questionnaire for the assessment of occupational well-being, the Questionnaire on the Experience and Evaluation of Work, then revised in 2015 and improved through the addition of other scales. The existence of this instrument clearly does not detract from the authors' work, but I think it would be appropriate to at least make mention of it in the literature review, especially in light of the statement in lines 42-43.

Response 1:

We really appreciated your constructive question and suggestion. In our revised manuscript, we have carefully followed your comments and revised all comments as suggested. First, we have added Veldhoven and Meijman (1994) in our scale measures. Second, we have modified and adopted in the review section (see literature review). Third, we have adopted some scale in our scale development and validation (see method section).

Point 2:

I found it quite difficult to understand well what the authors wrote due to the construction of the sentences in English; I therefore recommend a revision of the English for greater clarity.

Response 2:

Thank you very much for this constructive suggestion. Our manuscript has been proofread, and the quality of communication has been entirely improved by the Cambridge Proofreading LLC. The manuscript was edited for proper US English, grammar, punctuation, spelling, and overall style by two academic editors, who are a professional language editing our manuscript.

Reviewer 2 Report

Comments and Suggestions for Authors

Thank you for the opportunity to review this paper. Authors present the construction and development of a new psychometric tool to assess occupational well-being. The new psychometric instrument is a valuable addition to existing literature, as most of the already existing measures assess only some of the occupational well-being dimensions, while the newly developed tool assesses six dimensions and has very good psychometric characteristics. The methods are clear and coherent. However, there are some minor issues that need to be addressed.

Review: I think that the literature review is relevant and easy to follow. The gap in knowledge is identified and addressed.

Methods:

-        In the 3.4 chapter, it is not clear whether first authors generated a larger item pool (and how many items there were), and how these items were reduced to 47.

Discussion:

-        The “Discussion” part needs to be enriched. I expected to read more things regarding both the tool’s psychometric soundness and its practical usefulness.

-        It seems rather confusing the reference in Laos when stating the aim of the study. Is it an Occupational Well-Being assessment tool (that happened to be constructed in Laos and therefore the validation studies took place there), or a context-oriented Occupational Well-Being tool that is relevant only in Laos?

-        - I think some minor proofreading is needed (e.g. in 3.2. subchapter authors write “… scale developers should […] (3) avoid ambiguous sentences, (4) positively and negatively worded items [you mean avoid this kind of words?], (5) the same numbers [what do you mean?], (6) sensitive items [you mean avoid sensitive items?], and (7) levels of items [what do you mean?], and (8) be sure and write items [???]”)

Comments on the Quality of English Language

The use of English is good.

However, I think some minor proofreading is needed (e.g. in 3.2. subchapter authors write “… scale developers should […] (3) avoid ambiguous sentences, (4) positively and negatively worded items [you mean avoid this kind of words?], (5) the same numbers [what do you mean?], (6) sensitive items [you mean avoid sensitive items?], and (7) levels of items [what do you mean?], and (8) be sure and write items [???]”)

Author Response

The response to reviewer file is attached below.

Reviewer 3 Report

Comments and Suggestions for Authors

The article is promising in terms of the originality of the Occupational Well-Being measurement, as evidenced by the scale's properly conducted construction and validation process. However, to realize its full potential, the article should be edited; provide further evidence and arguments to support its relevance and an in-depth discussion.

The introductory section lacks coherence and structure. Excessive amounts of information compressed into a single paragraph can confuse readers, distracting them from the main concept of the article. To remedy this, authors should consider revising the introduction to provide a more compelling and engaging narrative that effectively introduces the main topic of the study. In addition, the introduction should strive to build a compelling foundation for the study, clearly identifying the gap in the existing literature and the importance of the study. The authors should more clearly emphasize their research objective and re-examine the literature to identify the validity of validating the GBS scale using just these following variables: affective, professional, social, cognitive, psychological, and psychosomatic well-being.

Moreover, the article aims to uniquely situate itself in the context of South East Asian conditions, which is Laos and employees working there. However, the article does not provide enough information from this context to make it relevant to the main concept of the study. To improve the contextualization of the article, the authors should include more specific examples or insights from the Lao organizational perspective that illustrate the relevance and uniqueness of the validation of the OWB scale under study in this specific environment.

In addition, the "Discussion of Key Results" section is quite limited. It should be significantly improved to highlight the results and their significance by explaining more clearly the new or valuable points, similarities and differences of the results compared to previous studies. I would recommend enriching the literature used in the manuscript. Thus, I would consistently recommend taking into account the consistency and readability issues that need to be resolved in order for the article to effectively communicate the research results.

Author Response

(The authors gave the same response as above.)
